# Tinnitus Guidelines and Their Evidence Base

**DOI:** 10.3390/jcm12093087

**Published:** 2023-04-24

**Authors:** Berthold Langguth, Tobias Kleinjung, Winfried Schlee, Sven Vanneste, Dirk De Ridder

**Affiliations:** 1Department of Psychiatry and Psychotherapy, University of Regensburg, 93053 Regensburg, Germany; 2Interdisciplinary Tinnitus Clinic, University of Regensburg, 93053 Regensburg, Germany; 3Department of Otorhinolaryngology, University Hospital Zurich, University of Zurich, 8091 Zurich, Switzerland; 4Institute for Information and Process Management, Eastern Switzerland University of Applied Sciences, 9001 St. Gallen, Switzerland; 5Trinity Institute for Neuroscience, Trinity College Dublin, D02 PN40 Dublin, Ireland; 6Global Brain Health Institute, Trinity College Dublin, D02 PN40 Dublin, Ireland; 7School of Psychology, Trinity College Dublin, D02 PN40 Dublin, Ireland; 8Section of Neurosurgery, Department of Surgical Sciences, Dunedin School of Medicine, University of Otago, Dunedin 9016, New Zealand

**Keywords:** tinnitus, evidence, meta-analysis, treatment guidelines, evidence-based medicine, living guideline, decision support system

## Abstract

Evidence-based medicine (EBM) is generally accepted as the gold standard for high-quality medicine and, thus, for managing patients with tinnitus. EBM integrates the best available scientific information with clinical experience and patient values to guide decision-making about clinical management. To help health care providers and clinicians, the available evidence is commonly translated into medical or clinical guidelines based on a consensus. These involve a systematic review of the literature and meta-analytic aggregation of research findings followed by the formulation of clinical recommendations. However, this approach also has limitations, which include a lack of consideration of individual patient characteristics, the susceptibility of guideline recommendations to material and immaterial conflicts of interest of guideline authors and long latencies till new knowledge is implemented in guidelines. A further important aspect in interpreting the existing literature is that the absence of evidence is not evidence of absence. These circumstances could result in the decoupling of recommendations and their supporting evidence, which becomes evident when guidelines from different countries differ in their recommendations. This opinion paper will discuss how these weaknesses can be addressed in tinnitus.

## 1. Introduction

### 1.1. Tinnitus and Evidence-Based Medicine

The treatment pathways of tinnitus patients vary largely from country to country and within countries. A tinnitus patient will be offered very different treatments depending on the institutions where the patient presents [1,2]. This indicates that for most patients, the treatment is not based on standards derived from scientific evidence but on standard-of–care-practices in the respective health system or experts’ recommendations, which are highly variable. The treatment may be biased on the clinician’s expertise depending on the health care provider. This is sometimes expressed as the ‘law of the instrument’ or ‘Maslow’s hammer’: “If the only tool you have is a hammer, it is tempting to treat everything as if it were a nail” [3]. This situation shows that the tinnitus field is clearly behind modern evidence-based medicine (EBM) standards, which will be analysed here. The term ‘evidence-based medicine’ was introduced as “the process of finding, appraising, and using contemporaneous research findings as the basis for medical decisions” [4] or as “the conscientious, explicit and judicious use of current best evidence in making decisions about the care of individual patients” [5]. In essence, evidence-based medicine integrates the best available scientific information with clinical experience and patient values to guide decision-making about clinical management.

To make it easier for clinicians and health care providers to apply the ever-growing body of available evidence judiciously, the evidence is commonly translated into medical or clinical guidelines by a guideline committee [6]. It differs from previous approaches, which go back to ancient times and were routinely based on tradition or authority. For example, the famous *Ebers papyrus*, dating back to 1550 BC, already had guidelines on treating tinnitus (strange ear) [7]. The transition from the results of clinical trials to clinical guidelines is a complex process. The first step is the search for all relevant published reports from clinical trials. The second step is the critical evaluation of the methodological quality of the trials. Then, in the third step, the results are aggregated. Among other criteria, this involves the evaluation of efficacy, effectiveness, side effects, validity, tolerability, feasibility and cost-effectiveness. Moreover, inconsistencies in clinical trial results and potential reasons for these inconsistencies (e.g., slight differences in the intervention, differences in study samples, insufficient statistical power, etc.) must be identified. Finally, there are standardized procedures for guideline committees to integrate all this information and develop recommendations for clinical management.

In the field of tinnitus, many different treatments have been offered to patients, which also reflects that no treatments are highly effective in all types of patients. With the advent of EBM, clinical researchers aimed to evaluate the effectiveness of various treatments by conducting clinical trials. This is far from trivial in tinnitus for several reasons [8,9,10]. First, tinnitus is a heterogeneous condition. Second, it is mostly purely subjective, which makes outcome measurement challenging, and third, suffering from tinnitus has many facets, which vary from patient to patient. All these aspects have been addressed in the last few years. Research has focused on identifying clinically relevant subtypes of tinnitus (ref). This was only partly successful, as the heterogeneity is better described by dimensional variability of tinnitus characteristics than by distinct categories (e.g., more or less somatic involvement instead of a category “somatic tinnitus”). Concerning the many facets of tinnitus burden, research aimed to identify consensus-based core outcome domains [11]. For outcome measurement, various tools have been developed and tested for psychometric adequacy. These efforts revealed that tinnitus questionnaires and visual analogue scales represent reliable and valid tools for quantifying tinnitus impact. In contrast, psychometric measurements of tinnitus loudness are not helpful for this purpose [12,13,14].

Moreover, many efforts have been made to establish methodological standards for patient assessment, outcome measurement and clinical trial methodology in tinnitus [11,14,15,16]. As a result, more clinical trials evaluated the effectiveness of the various tinnitus treatments (see Figure 1, Box 1). These studies enable systematic reviews and meta-analyses based on randomized controlled trials (RCTs), considered the strongest scientific evidence for creating clinical management guidelines.

Box 1Therapeutic interventions for tinnitus evaluated with randomized controlled trials (listed in alphabetical order)
**Pharmacological interventions.**

*Antidepressants*

*Amitriptyline*

*Nortriptyline*

*Paroxetine*

*Sertraline*

*Trimipramine*

*Anticonvulsants*

*Carbamazepine*

*Gabapentin*

*Lamotrigine*

*Selurampanel*

*Benzodiazepines/GABAergic drugs*

*Alprazolam*

*Baclofen*

*Clonazepam*

*Diazepam*

*Glutamatergic drugs*

*Acamprosate*

*Esketamine*

*Memantine*

*Neremexane*

*Muscle relaxants*

*Cyclobenzaprine*

*Eperisone*

*Orphenadrine*

*Tizanidine*

*Sodium channel blocker*

*Lidocaine*

*Others*

*Atorvastatin*

*Betahistine*

*Chinese medicine*

*Cilostazol*

*Cyclandelate*

*Deanxit*

*Ginkgo biloba*

*Melatonin*

*Misoprostol*

*3,4-Methylenedioxymethamphetamine (MDMA)*

*Naloxone*

*Odansetron*

*Oxytocin*

*Piribedil*

*Pramipexole*

*Vardenafil*

*Vitamin B12*

*Zinc*

**
*Non-pharmacological interventions:*
**

*Acupuncture/Acupressure*

*Auditory Training*

*Bimodal stimulation*


*Vagus nerve stimulation plus sound therapy*

*Electrical skin stimulation plus sound therapy*

*Electrical tongue stimulation plus sound therapy*


*Brain/neural stimulation*


*Transcranial magnetic stimulation*

*Transcranial direct current stimulation*

*Direct electrical stimulation*

*Vagus nerve stimulation*

*Transcutaneous electrical neural stimulation*


*Combination Approaches*


*Tinnitus Retraining Therapy (directive counselling plus sound therapy)*

*Neuromonics (counselling plus acoustic stimulation)*


*Electrical stimulation of the ear/cochlea*


*Cochlear implants*

*Electrical stimulation of the tympanum or the outer ear canal*


*Hearing Aids*

*Hyperbaric Oxygenation*

*Low-Level Laser Therapy*

*Music Therapy*

*Neurofeedback*

*Physiotherapy*

*Psychotherapy*


*Cognitive behavioural therapy (group setting)*

*Cognitive behavioural therapy (individual setting)*

*Online/internet based Cognitive behavioural therapy*

*Mindfulness-based therapy*

*Hypnosis*

*Virtual Reality based approaches*


*Sound Treatment*


*Noise generator (complete masking)*

*Noise generator (partial masking)*

*Enriched acoustic environment*

*Fractal Tones*

*Taylor-made notched music training*

*Coordinated reset auditory stimulation*



Here we will discuss the translation of evidence into clinical guidelines in general and for the tinnitus field in particular. We will shortly review the current evidence and then focus on the challenges of guideline development in the tinnitus field and approaches how to address them.

### 1.2. Translating Evidence from Clinical Trials into Guideline Recommendations

The first step is defining the clinical question, the relevant patient population, and the relevant outcomes. Based on a systematic literature search, the appropriate studies and systematic reviews of these are identified. Subsequently, the quality of evidence for all relevant outcomes is assessed. Systematic approaches to evaluate the level of evidence and the certainty of evidence have been developed. The certainty of evidence depends on the type of clinical trial and the methodological rigor of the trial. An example of a classification system of clinical trials according to the level of evidence is given in Table 1. Other classification systems differ slightly in the criteria.

Evidence from randomized controlled trials starts at high quality, and evidence that includes observational data starts at low quality. The certainty in the evidence is increased or decreased by one or two levels according to further study criteria, resulting in a final level of certainty rating (see Figure 2).

With the GRADE system (Grading of Recommendations, Assessment, Development and Evaluations), a transparent framework has been proposed for creating clinical practice recommendations. GRADE has four levels of evidence—also known as certainty in evidence or quality of evidence (Table 2). The main criterion for reducing the level of certainty is the risk of bias. *Bias* occurs when the results of a study do not represent the truth because of inherent limitations in the design or conduct of a study. Risks of bias include *sample selection bias* (the sample is not representative of the patient population, *publication bias* (negative results are more likely not to be published) or *funding bias* (influencing the study design by a funder to achieve a particular outcome). Certainty in a body of evidence is highest when several studies show consistent effects. Conversely, certainty may be downgraded if the patients studied differ from those to whom the recommendation applies. The same is true if the setting of a study is different from the real-world condition (for example, a study of a psychotherapeutic approach by a specialized therapist only indirectly applies to a psychotherapist with less experience).

In some circumstances, certainty in the evidence can be rated up (see Figure 2) when there is a large magnitude of effect and, second, when there is a clear dose-response gradient. Third, when residual confounding is likely to decrease rather than increase the magnitude of the effect (in situations where there is an effect). In GRADE, recommendations can be strong or weak, in favor or against an intervention. Besides the level of evidence and the certainty of the evidence, other factors play a role. The balance between the desirable consequences and the adverse effects of a given treatment is very important. In addition, patient values and preferences should be considered. For example, in tinnitus, patients’ expectations of their treatment vary widely. If a patient expects a reduction of tinnitus loudness, a treatment effective for reducing tinnitus suffering might not satisfy the patient. Finally, the availability of a given treatment and the required resources should also be considered. Thus, if a low-cost and widely available treatment has the same effectiveness as a costly treatment only available in specialized centers, the recommendation should favor the widely available low-cost treatment.

A fair and balanced comparative evaluation of the efficacy and safety of various therapeutic interventions is challenging as the criteria for grading the evidence level are not equally applicable to different interventions. This can be best illustrated by the requirement of a control condition with a double-blinded group allocation of study participants. This may be feasible for a pharmacological intervention but impossible for testing a hearing device, counselling, or cognitive behavioral therapy. When testing these latter interventions, compromises in methodological quality must be accepted. However, comparing a certain intervention with a waiting list control group (ref) may favor this intervention when results are compared to an intervention tested in a placebo-controlled trial with effective blinding. Similar considerations are true for the assessments of the safety of therapeutic interventions. Whereas a thorough safety assessment is mandatory in pharmacological trials, side effects are only incompletely documented in trials that evaluate psychotherapeutic interventions [17].

Moreover, the extent of tinnitus burden is strongly influenced by co-morbidities such as hearing loss, depression or anxiety [18]. Interventions that act primarily on these co-morbidities (e.g., hearing aids or antidepressants) may benefit the tinnitus patient. Still, the specific effect on tinnitus might be difficult to disentangle from the beneficial effect on the co-morbidity. When integrating all these aspects, it becomes clear that there is a certain margin of discretion in translating evidence from clinical trials into clinical recommendations. As the weighting of the various aspects is highly subjective, the composition of the guideline committee and their potential conflicts of interest gain relevance.

In this perspective paper, we aim to discuss the strengths and weaknesses of the current adaption of evidence-based medicine in tinnitus. For this purpose, we will summarize the currently existing guidelines based on a literature search and contrast them with each other and with the currently available evidence.

## 2. Materials and Methods

As a first step, we aimed to identify published guidelines for tinnitus management. For this purpose, we have performed systematic literature research with the keywords “tinnitus” and “guidelines” in PubMed and Google, with the last update on 1 December 2022. To identify the evidence basis on which the guideline is based, we performed additional systematic literature research with the keywords “meta-analysis”, “tinnitus”, “Cochrane”, and “randomized controlled trials” in PubMed with the last update 1 December 2022. We then followed a hierarchical approach according to the established levels of evidence for therapeutic studies (see Table 1). If there existed a recent Cochrane Meta-analysis for a given intervention, we chose this Cochrane Meta-analysis as the evidence base for this intervention. If there was no Cochrane Meta-analysis for a given intervention, we looked for systematic reviews or meta-analyses of randomized controlled trials and used these. Finally, we looked for RCTs if there was no systematic review or meta-analysis. It is essential to state that we never intended to provide an overview of all treatment studies in the tinnitus field. Our methodological approach was driven by the intention to contrast the existing tinnitus guidelines with each other and with the available evidence basis for the various interventions mentioned in the guidelines.

## 3. Results

### 3.1. Overview of the Evidence for Therapeutic Interventions in Tinnitus

As mentioned above, many different tinnitus treatments have been evaluated in clinical trials. Box 1 provides a (possibly incomplete) overview of therapeutic interventions investigated in clinical trials. Notably, not all these investigated interventions are revised in the guidelines.

In Table 3, we give a synoptic overview of the most widely investigated interventions, their evidence base and the respective recommendations in the various guidelines. In addition, we provide a short narrative overview of the most commonly used therapeutic interventions.

#### 3.1.1. Tinnitus Counselling

Tinnitus Counselling is typically considered a basic therapeutic approach. It is recommended by all guidelines, even if the evidence for the efficacy of randomized controlled trials is limited. Educational counselling was used as an active control condition in several studies, where it had a beneficial effect [38,39]. Additional studies investigating counselling through specific smartphone apps are underway [40].

#### 3.1.2. Cognitive Behavioral Therapy for Tinnitus

Cognitive behavioural therapy is the best-investigated treatment intervention for tinnitus. The main findings from a recent Cochrane meta-analysis [17] are that cognitive behavioural therapy (CBT) can effectively reduce the score of tinnitus questionnaires at the end of treatment and that there are few if any, adverse effects from receiving CBT (although further research on this is recommended). It is unclear how long the treatment effects last, as there are only a few 6- or 12-month follow-up data [17]. CBT for tinnitus may also reduce symptoms of depression and improve anxiety, health-related quality of life or negatively biased interpretations of tinnitus. Still, the strength of evidence for these effects is low. CBT delivered individually, group-wise and via the Internet, with some additional email communication from a professional, appear similarly effective. It should be noted that only a subgroup of patients is willing to undergo CBT and that the availability of tinnitus-specific CBT is limited. Innovative forms of CBT, such as smartphone App based CBT or virtual reality-based CBT [41], might enhance its acceptance and frequency of use.

#### 3.1.3. Mindfulness and Tinnitus

Mindfulness-based treatments are sometimes subsumed under CBT and included in systematic reviews of CBT. A recent systematic review focusing exclusively on mindfulness-based interventions (MBI) [42] concludes that MBI decreases tinnitus distress scores directly post-therapy based on moderate to high-quality studies. This was found regardless of the heterogeneity of patients, study design, type of MBI and outcome assessment. Two out of three RCTs found clinically relevant decreases in mean tinnitus distress scores. No effect of MBIs was observed for depression and anxiety in tinnitus patients. Long-term effects remain uncertain.

#### 3.1.4. Virtual Reality-Based Treatment

In a large randomized clinical trial, patients were randomized into either a virtual reality (VR) based intervention or CBT, with both groups demonstrating similar improvement [41]. These findings suggest the potential of VR-based interventions and warrant further research.

#### 3.1.5. Auditory Treatments of Tinnitus

Auditory treatments can be divided into four categories: devices for improving hearing, sound generators for tinnitus masking, auditory stimulation to induce specific neuroplastic changes in the central auditory system and auditory training.

Concerning the improvement of hearing, there is meta-analytic evidence for the efficacy of cochlear implants on tinnitus in patients with unilateral severe hearing loss or deafness [29,43]. Even if data come mainly from observational studies and not from RCTs, the effect size of 1.32 is clinically highly relevant. Concerning hearing aids, the evidence from randomized controlled trials is very limited [32]. A recent meta-analysis has shown that the results strongly depend on the procedure and how hearing aid fitting is performed [44]. The relevant question of whether people with tinnitus who have hearing loss but do not have difficulties communicating may benefit from hearing aids has not been addressed in RCTs. The evidence for sound generators is very limited as well [32]. Moreover, concerns were raised that chronic exposition to white noise may have a harmful effect on hearing [45].

Some forms of auditory stimulation, such as enriched acoustic environment [46,47], amplitude-modulated tones [48,49,50], tailor-made notched music training [51,52] or coordinated reset auditory stimulation [53] aim to reduce tinnitus by the induction of specific neuroplastic changes in the central auditory system. However, only data from pilot studies are available for all these approaches, which require confirmation in larger RCTs before recommendations can be made.

Auditory training approaches encompass various training procedures, e.g., improving frequency discrimination, sound localization or signal in noise detection. In a systematic review, the quality of the studies investigating auditory training was rated low, and the need for appropriate RCTs was expressed [27].

#### 3.1.6. Tinnitus Retraining Therapy

Tinnitus Retraining Therapy (TRT) consists of a combination of counselling and sound therapy. According to a recent meta-analysis [34], TRT as an add-on to standard treatment improved the response rate after one month, three months and six months. However, the quality of evidence of the available studies has been rated as low, with a high risk of bias. Moreover, there is an ongoing debate about the contribution of sound therapy to TRT effects. However, according to a recent multicentric study, TRT with and without sound therapy did not differ in their effects [54].

#### 3.1.7. Tinnitus Pharmacotherapy

Various pharmacological agents have been investigated, and meta-analyses have been performed for several compounds. Most of these studies were negative. Exceptions are some experimental approaches, which cannot be performed as routine treatments because of side effects (Lidocaine), and some positive pilot studies [55,56,57], which could not yet be replicated by large confirmatory RCTs. Cochrane meta-analyses for Ginkgo biloba [31,58], anticonvulsants [25] and antidepressants [26] were negative. There is some limited evidence for pharmacological treatment of comorbid conditions: Antidepressants have shown beneficial effects for comorbid depressive symptoms [59], melatonin improved sleep in tinnitus patients in controlled studies [60,61,62], and Ginkgo biloba was shown to be beneficial in elderly tinnitus patients with dementia [53].

#### 3.1.8. Tinnitus Activities Treatment

Tinnitus activities treatment has not been investigated in controlled trials. There exists only one controlled trial in which tinnitus activities were the basis of treatment, and patients were randomized to additional sound therapy [63].

#### 3.1.9. Neural Therapy and Botox

Only pilot studies are available for neural therapy and botox [64,65,66], which are insufficient for any recommendation.

#### 3.1.10. Physiotherapy

Different forms of physiotherapy have been investigated with promising results. Most of these studies focused on patients with tinnitus and comorbid temporomandibular joint or neck disorders. Large-scale randomized controlled trials are needed to determine which interventions are effective in which patient groups [67].

#### 3.1.11. Neurobiofeedback

Different neurobiofeedback paradigms have been investigated in several controlled studies, with promising results [68]. However, large confirmatory studies are needed before recommendations can be made.

#### 3.1.12. Non-Invasive Brain Stimulation

Meta-analyses reveal positive effects for both transcranial direct current stimulation (tDCS) [35] and repetitive transcranial magnetic stimulation (rTMS) [36] with a small to medium effect size (between 0.35 and 0.5). For rTMS, an effect size in this range (0.42) has also been observed at follow-up assessments one week to six months after treatment. A recent network meta-analysis investigated the various investigated stimulation protocols of tDCS and rTMS separately, which resulted in effect sizes between -1.89 and 0.11, illustrating the relevance of the specific stimulation protocols [69].

#### 3.1.13. Invasive Brain Stimulation

Invasive brain stimulation represents a highly experimental treatment requiring surgical insertion of electrodes under the skull. Epidural, subdural and deep brain stimulation has shown to be beneficial in case series, but there is far from sufficient data to support their routine clinical use [70].

#### 3.1.14. Bimodal Stimulation for the Treatment of Tinnitus

Different forms of bimodal stimulation have been investigated in the last years in randomized controlled trials. One involves the presentation of tones in combination with vagal nerve stimulation via an implanted vagus nerve stimulation device [71]. Other approaches combined auditory stimulation with electrical stimulation of the face or neck area [72] or the tongue [73,74]. The approaches differ slightly in using auditory stimuli and the timing between auditory and electrical stimulation. The first studies have shown substantial reductions in tinnitus severity by all three approaches in studies with 20 [72], 30 [71], 326 [73] and 191 [74] participants.

#### 3.1.15. Complementary and Alternative Therapies

Of the many complementary and alternative therapies, only a few have been investigated in RCTs. The best-studied among these treatments is probably acupuncture, for which a recent systematic review identified 8 RCTs with 504 participants. Acupuncture had no significant effect on the primary outcome of the VAS score compared with control treatment but positive effects on secondary outcomes (THI and TSI score). However, the authors conclude that due to the low quality and small sample size of the included trials, the level of evidence is insufficient to draw any definitive conclusions [75].

#### 3.1.16. E-Health Based Approach

E-Health-based approaches involve online and offline-based treatments, including web-based– and smartphone apps. In particular, the number of smartphone apps for tinnitus management is rapidly growing [76], although the evidence for their efficacy is still very limited. On the other hand, a relatively large number of studies investigating CBT for tinnitus in a blended approach combining face-to-face and internet-based applications reveals that this approach is similarly effective as CBT delivered individually and group-wise [17].

#### 3.1.17. Self-Help Interventions

The term “self-help intervention” is used in different contexts with different meanings. Whereas “self-help” traditionally means the mutual exchange of patients in self-help groups, the term “self-help” has been recently used in online-delivered CBT. A qualitative analysis of the traditional self-help concept revealed that members of self-help groups see the importance of their group primarily in social inclusion, psychosocial relief, coping with the disease and new insights into dealing with the disease. A systematic review of the traditional self-help approach concluded that because of the lack of high-quality and homogeneous studies, no confident conclusions could be drawn regarding the efficacy of self-help interventions for tinnitus [77].

The evidence of the most widely used and investigated interventions and the respective recommendations in the various guidelines are summarized in Table 3.

### 3.2. Guidelines Overview

There exist several guidelines for the management of tinnitus [78]. They vary largely in scope and extent, in their methodological rigor, organisational structure, actuality, and composition of the guideline committees. The various guidelines are presented in more detail below.

#### 3.2.1. NICE Guidelines (UK)

The NICE Guidelines [79] were compiled in 2020 by the British National Institute for Health and Care Excellence (NICE) according to the highest methodological rigor. The guideline development process is highly standardized and transparent [22]. The evidence used to develop the recommendations is summarized in 15 evidence reports, comprising up to 250 pages each [22]. The recommendations of the NICE guidelines differ in several important aspects from other guidelines. One example is the recommendation of hearing aids. The NICE guideline differentiates between tinnitus patients with hearing loss that affects their ability to communicate and those with hearing loss but no communication difficulties. This differentiation is essential, as in the first group, hearing aids are already indicated for improving communication. In contrast, in the second group, the indication for the hearing aid is primarily its potentially beneficial effect on tinnitus. This differentiation results in a graded recommendation: amplification devices should be offered to the first group and can be considered in the second group. Another example is the recommendation for CBT, which considers availability and resources in addition to effectiveness. Thus, the guideline proposes a stepwise approach, starting with digital tinnitus-related CBT, offering group-based psychological interventions (mindfulness-based cognitive therapy, acceptance and commitment therapy or CBT) as the second step and individual CBT as the third step. In contrast to other guidelines, the NICE guidelines make recommendations for or against specific interventions, but also “recommendations for “research”. Such a recommendation is given when evidence for the efficacy or safety of a specific intervention is inconclusive or insufficient.

#### 3.2.2. German Guideline

The German guideline was edited in 2021 by the German Society of Ear-, Nose- and Throat-medicine as an S3 Guideline [23], which means that the guideline has formally undergone all elements of systematic development (logic, decision and outcome analysis, assessment of clinical relevance of scientific studies and regular review). The evidence on which the guideline is based is summarized in evidence tables in the appendix of the guideline. Recommendations for and against are provided in 3 categories each (strong, recommendation, no recommendation). For some recommendations, the German guideline differs from all other guidelines, e.g., in the recommendation for auditory training or the recommendation against sound therapy (see Table 3).

#### 3.2.3. Clinical Practice Guideline: Tinnitus (US)

The American Academy of Otolaryngology—Head and Neck Surgery Foundation (AAO-HNSF) developed and published the US guideline in 2014 [19]. I use an explicit and transparent a priori protocol for creating actionable statements based on supporting evidence and the associated balance of benefit and harm. Recommendations for and against are provided in 4 categories each (strong recommendation, recommendation, option, no recommendation). The US Guideline differs from the other guidelines in the case of sound therapy, which is mentioned as a treatment option only in the US guidelines. A major limitation of the US guideline is its compilation date and the lack of an update.

#### 3.2.4. Swiss Guideline

The Swiss Guideline is a very short practice guideline, edited in 2019 by an association of networks of medical practitioners (Medix) [20]. It is neither backed by a medical society nor by experts in the field, and there is no information about the methodology of its development. The guideline does not provide recommendations but a very short summary of the available evidence for various therapeutic interventions.

#### 3.2.5. European Guideline

The European Guideline has not been edited by society but by a group of clinicians and researchers from different European countries who participated in the EU-funded COST Action TINNET [21]. Neither the procedure for collecting evidence nor the method used to transition from evidence to clinical recommendations has been described in detail. There is no summary of the evidence on which the guideline has been based. Recommendations for and against are provided in 4 categories each (strong recommendation, recommendation, weak recommendation, no recommendation).

#### 3.2.6. Japanese Guideline

The Japanese Clinical Practice Guideline for Diagnosis and Treatment of Chronic Tinnitus was developed by subcommittee members, edited by the Japan Audiological Society, authorized by the Oto-Rhino-Laryngological Society of Japan and published in 2019 [24]. Recommendations are graded in strong recommendation [1], recommendation [2] and no recommendation (none). The evidence level of each recommendation is graded in 4 categories (A, B, C, D) and is added to each recommendation.

## 4. Discussion

We will first discuss the translation of the currently available clinical evidence into the different guidelines. In the second part, we will provide suggestions for further development.

### 4.1. Current Evidence in Tinnitus Treatment: Limitations and Challenges

As mentioned above, the various existing guidelines differ in several aspects. This has many reasons. First, the guidelines differ in actuality and, thus, in the body of evidence based on their conclusions. Second, the methodological rigor, the systematics, and the transparency of how evidence was collected from the literature vary largely across guidelines. Third, the transition from evidence to recommendations is a process which involves subjective evaluations, considerations and balances and thus depends on the constitution of the guideline committee with their individual biases and conflicts of interest. All these factors may explain the variable interpretation of the evidence by different guidelines, which results in substantial variability across guidelines (see Table 3). Sound treatment, for example, is optionally recommended by the US guideline, whereas the German guideline recommends against sound generators and specific sound therapies. Some treatments are only recommended by one guideline and not be others (e.g., auditory training in the German guideline). There is also a considerable difference in the recommendations in case of inconclusive evidence. While most guidelines recommend against such situations, the NICE guidelines emphasize the need for further research. An aspect being considered is also the categorization of recommendations. CBT is strongly recommended by various guidelines based on an effect size of −0.56 immediately after treatment with low certainty of evidence and an absence of evidence at 6 or 12 months of follow-up [17]. A meta-analysis for transcranial magnetic stimulation reveals an effect size of −0.45 immediately after treatment and −0.42 at follow-up (between 1 week and six months after treatment) [36], but it is not recommended by any guideline. Thus, a relatively small difference in the evidence results in different recommendations. Put in another way, by looking at similar outcome scores and based on the meta-analysis CBT reduces tinnitus distress with 10.91 points on the THI scale, which is a scale from 0–100. This is minimally more than the 7-point improvement required to reach the MCID [80] and much less than the effect of cochlear implants, which improve the THI by 23.2 points [81]. However, the mean THI reduction from CBT is similar to the meta-analytic findings in rTMS (THI: −7 to −8) [36,82,83], tDCS (THI: −9.69) [84] and acupuncture (THI: −8.28 to −10) [75,85]. Yet, all guidelines will recommend CBT, but not the other treatments with similar meta-analytic effect sizes. The NICE guidelines explain the different assessments of the evidence in this case by stating, “While the committee were able to make recommendations for other [treatments] based on very limited evidence, they decided that because neuromodulation interventions are currently not offered for tinnitus on the NHS, any recommendation would have a large impact on current practice and there was therefore not enough evidence to support this change” [22]. This implies that the recommendation for new and innovative treatments requires another threshold of evidence as compared to treatments that are in routine use. This aspect needs further consideration as it impacts the development of innovative treatments urgently needed in the tinnitus field. According to the statement of NICE (which is probably the most innovation-friendly guideline, as it provides “recommendations for research” and not “recommendations against” if the evidence for an innovative treatment is not yet sufficient), a new treatment would require high-level evidence for superior efficacy and safety as compared to the currently recommended treatments. Thus, the threshold for recommendation of an innovative treatment in the guidelines is higher than the requirements for FDA approval in the USA, where a non-inferiority trial is sufficient. The FDA approach is based on a model where new and established treatments should be treated equally. Furthermore, to demonstrate non-inferiority, a smaller (less expensive) study can suffice compared to a study powered for superiority.

There is a trade-off between the certainty of evidence (that can be higher in established treatments, which have been investigated in many clinical trials) and the openness to innovation (as the amount of clinical data is typically lower for new treatments). However, with guideline update cycles of at least five years, it would take about ten years under optimal conditions for a new safe and effective treatment to be recommended in the guidelines. Furthermore, as a recommendation in guidelines plays a major role in the implementation of clinical practice and payment by healthcare providers, the question arises whether it can be justified that patients have to wait so long until they can benefit from an innovative therapy. Furthermore, from an economic perspective, earlier availability of effective, innovative treatments can reduce tinnitus’s enormous direct and indirect socio-economic costs [86,87,88]. Finally, a delay of 10+ years from a successfully developed innovative treatment till its recommendation in guidelines will limit the willingness to invest in innovative tinnitus treatments.

Undoubtedly, the methodology of evidence-based medicine (EBM) has improved the quality of medical services immensely. But EBM also has its inherent limitations and weaknesses. The methodological gold standard—studying a therapeutic intervention in many multi-centric randomized controlled trials and performing a meta-analysis of their results—aims to focus on the essential aspect of the pathological condition and the therapeutic intervention by removing all patient- and therapist-specific aspects by averaging across many patients and treatment settings. The more homogeneous the pathophysiology of a given disorder and the more homogeneous the intervention, the more appropriate this approach is. However, in the case of a heterogeneous condition such as tinnitus, this averaging approach has major limitations. Therefore, it is of utmost importance to identify criteria for meaningful tinnitus subtypes, as this would allow the investigation of specific interventions for these specific subtypes. The same applies to the heterogeneity of interventions. In neurostimulation, the investigated interventions vary in many parameters [89]. Therefore, pooling all these studies in meta-analyses may not be the most appropriate approach. Recently, the method of network meta-analyses was developed, which enables the aggregated analysis of various interventions. Network meta-analyses for neurostimulation [69,90] in tinnitus, considering the differences between the different stimulation protocols, suggest more promising results than the typical standard meta-analysis. However, these network meta-analyses were not yet considered in developing guidelines. A further important aspect is the transferability of results from clinical trials into clinical practice. In the case of administering a medication, it may be guaranteed that the patients in the real world receive the same treatment as the patients in the clinical trials. However, for example, there might be differences between the treatment provided by the resident psychotherapist with limited tinnitus experience in cognitive behavioural therapy and the specialized therapist in a tertiary referral center where treatment studies are performed.

Finally, it is remarkable that we could only identify guidelines from European countries, Japan and the US. This means that no guidelines are available for large parts of the world. Furthermore, as tinnitus suffering and coping also involve cultural aspects, it is questionable whether recommendations from Europe, Japan or the US can be applied to patients in other parts of the world.

### 4.2. Considerations for Future Directions

#### 4.2.1. Who Should Write the Guideline?

Although evidence-based medicine and the development of guidelines have made an important contribution to a scientific approach in medicine and the tinnitus field, one should be aware of their limitations and seek strategies to address them. As mentioned above, the guidelines committees, who translate the evidence from clinical trials into clinical recommendations, have a considerable margin of discretion. This aspect is particularly relevant in a highly multidisciplinary field like tinnitus, where audiologists, ENT specialists, general practitioners, neurologists, neurosurgeons, psychiatrists, psychologists and physiotherapists treat patients. Clinicians from different disciplines might be more familiar with the interventions in their discipline and might have an interest in placing them prominently in the guidelines. For example, a psychotherapist will have mainly experience with psychological treatment and be more in favour of it, whereas an audiologist is mainly familiar with sound-based interventions. Therefore, potential conflicts of interest of the guideline committee members are of utmost importance. In practice, the composition of guideline committees varies considerably.

For some guidelines, the composition follows specific rules (e.g., German Guidelines). In other guidelines, the committee comprises interested researchers or clinicians (e.g., European Guidelines). Finally, in some guidelines, it is unclear how the committee has been composed (e.g., Swiss Guidelines). In the case of the NICE, the guidelines have been developed by a team from an institute specialized in quality assessment of clinical evidence. Typically, all committee members have to declare potential conflicts of interest. This helps to disclose financial or material interests. However, it becomes more difficult concerning immaterial conflicts of interest.

Moreover, in most cases, there is no sufficient funding for guideline development. This might lead to the situation that guideline development is only possible by the voluntary commitment of experienced clinicians and researchers, who might not be experts in evidence-based medicine and may have an increased risk for material or immaterial conflicts of interest. In this context, sufficient public funding for guideline development is essential. Such funding enables the assessment of clinical evidence by an independent committee with experience in quality assessment, as is the case with the NICE guidelines. The lack of tinnitus-specific experience of such a committee might be compensated by inviting expert consultants in the guideline process. Whereas specialists in evidence assessment can ensure consistent criteria are applied for evaluating clinical studies of the various interventions, experts with tinnitus experience are needed to translate evidence to clinical recommendations. Independently of the composition of the guideline committee, all up-gradings and down-gradings of evidence (see Figure 2) must be transparently justified and should be applied consistently across various treatments.

#### 4.2.2. How to Address the Trade-Off between Experience and Innovation?

As mentioned above, there is a certain trade-off between recommendations based on a huge amount of evidence collected over many years and the openness to innovation. Guideline committees should know this trade-off and consider that the evidence, especially for long-term outcomes, is naturally limited to innovative interventions. In such situations, the potential benefit of the innovation must be weighed against a certain amount of insecurity. Such situations might require adjustments of the criteria and thresholds for recommendation, as was the case in recommendations for newly developed mRNA vaccines during the SARS-2 COVID-19 pandemic. On the other hand, suppose guideline committees realize the potential of an innovative intervention but consider the available evidence for safety and efficacy as insufficient. In that case, recommendations for further research with specifications about the additionally needed information (e.g., more safety data) is a much more constructive and innovation-friendly approach than just recommending against an innovative intervention. In this context, it is of utmost importance to stress that the patient’s interests should stand in the foreground and not the interests of the advocates for the established treatments, frequently constituting the guideline committees.

#### 4.2.3. How Can Guidelines Become More Up-to-Date?

An important limitation of the guideline process is the long delay between the publication of research data and their consideration in the guidelines. For example, the most recent US tinnitus guideline was published in 2014 and based on literature research from the spring of 2013. About half of the pub-med hits for “tinnitus randomized controlled trials” were published in 2013. This means the newer half of the relevant literature is not considered in the “actual” guideline. During the SARS-2 COVID-19 pandemic, the medical field has witnessed how fast medical progress can happen if needed. The living guideline process has been developed to cope with a particularly rapid increase in knowledge [91]. This process involves extracting data from relevant studies immediately after publication, their deposit in databases and analysis workflows, which enable an always up-to-date systematic literature review. This is combined with evaluating the guideline panel and leadership group in short cycles, up to weekly actualisations [92] (Figure 3).

Such living guidelines, actualised in short cycles, also require innovative dissemination and communication methods, e.g., mobile app-based guideline versions [93] with sophisticated search functions complemented by decision support systems [94,95] and instructive graphical design. We are aware that the development of new knowledge in tinnitus is not occurring at a speed that would require weekly guideline updates. Still, yearly actualisation cycles would be highly desirable.

#### 4.2.4. What Can Contribute to Better Evidence in Tinnitus Treatment?

A significant problem of the tinnitus field is the relatively low evidence level for most interventions. This is due to a lack of clinical trials with high methodological standards and small effect sizes for the various interventions. Many attempts have been made during the last decades to improve clinical trial methodology [15,94,96,97,98] and develop valid and standardized outcome measurements [11,14,16,99,100,101]. A further need is the identification of meaningful subgroups of tinnitus. There is consensus in the field that tinnitus is highly heterogeneous with respect to its aetiology, its phenomenology, and its pathophysiological mechanisms [102,103]. Moreover, it is assumed that subtypes of tinnitus, which differ in their pathophysiological mechanisms, presumably also differ in their response to specific therapeutic interventions. However, further research is needed to identify meaningful criteria for subtypisation.

#### 4.2.5. Who Should Treat Tinnitus Patients?

Another aspect is the multidisciplinary nature of tinnitus management. To prevent a Maslow Hammer effect, in guideline committees and clinical management, it may be optimal to develop multidisciplinary tinnitus centers, where the cumulative knowledge of the different healthcare providers can be offered to the patient. A multidisciplinary tinnitus center where audiologists, psychologists, ENT specialists, general practitioners, neurologists, psychiatrists, neurosurgeons and physiotherapists can provide their expertise and evidence-based knowledge may be superior to a single specialist working in isolation. This may become ever more essential once subtyping makes the clinical management of the tinnitus patient even more complex. In situations where a multidisciplinary tinnitus clinic is not possible, developing a multidisciplinary network of practitioners can fulfil the same needs.

#### 4.2.6. Involving Patients in Guideline Recommendations

Considering that the final goal of guidelines is a better treatment of patients with tinnitus, it is essential to understand patient values better, wishes and goals related to their clinical management. Even though organisations that develop clinical practice guidelines encourage the involvement of patients and the public in their development, there are no standard methodologies for doing so [104]. A trial could be performed to develop guidelines based on evidence-based medicine and clinical experience, with and without patient involvement, and verify whether these guidelines would be similar. Yet, analogous to the different biases introduced by different disciplines in tinnitus management, involving patients may introduce an even bigger bias depending on the different personality types of patients. On the other hand, all guidelines’ goals are optimal patient care and who could better advocate for this goal than the people affected by tinnitus.

## 5. Conclusions

Currently, tinnitus management varies widely across countries, disciplines, and institutions. In this context, developing treatment guidelines based on available evidence represents an important step towards a standardized treatment approach. We reviewed the evidence for various treatments and current tinnitus guidelines and concluded that the evidence-based treatment options are limited and, in many cases, unsatisfactory. Therefore, there is an urgent need to develop newer and better treatments for tinnitus. Guidelines are, by definition, conservative, as there is naturally more experience and, in most cases, more evidence for established treatments than innovative approaches. On the other hand, guidelines should not be hostile towards innovations. Here we propose strategies how to address this dilemma: (1) perform studies at higher methodological standards under consideration of the heterogeneity of tinnitus, (2) avoid a bias of the guideline committees towards established treatments, (3) formulate research needs instead of recommendation against intervention in case of insufficient data, (4) faster update cycles according to the living guideline concept and (5) more intensive involvement of tinnitus patients in guideline development. We are convinced that considering these aspects will make it possible to increase the quality standards of tinnitus management through evidence-based guidelines and, at the same time, create an innovation-friendly environment.

Finally, we should not forget that the ultimate goal of the healthcare provider is to help the tinnitus patient. Guidelines are not ‘laws’, merely based on as much evidence as we currently have. The clinician treats individual patients and always has the therapeutic freedom to offer off-label treatments if their use can be justified. This option should not be dismissed if the quality of evidence is low for most current tinnitus treatments. For the individual case, chances of improvement, risks of treatment, alternative options, overall health state, psychosocial situation and the patient’s values and desires must be weighted in a holistic approach to make a reasonable clinical decision.

## Figures and Tables

**Figure 1 jcm-12-03087-f001:**
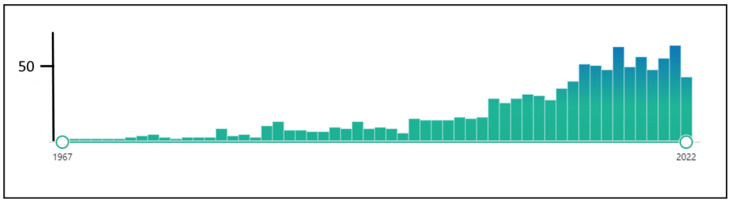
The number of hits for “randomized controlled trials tinnitus” in Pubmed (assessed 1 December 2022).

**Figure 2 jcm-12-03087-f002:**
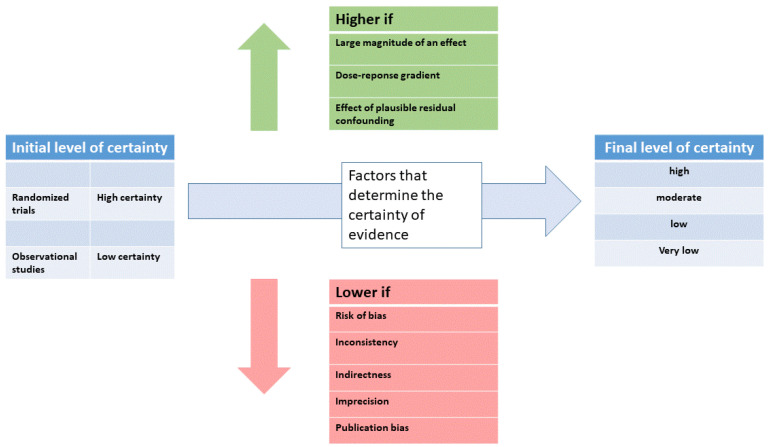
Schematic illustration of the procedure to determine the level of certainty (modified from [6]).

**Figure 3 jcm-12-03087-f003:**
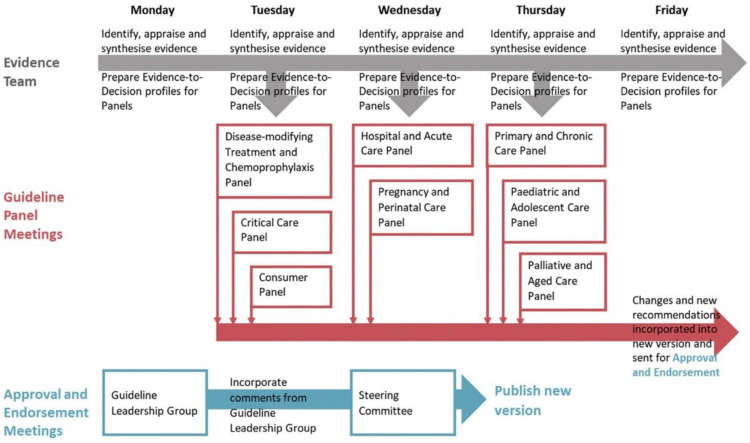
The weekly flow of evidence, recommendations, and approval (from [92]).

**Table 1 jcm-12-03087-t001:** Levels of Evidence for Therapeutic Studies (From the Centre for Evidence-Based Medicine, http://www.cebm.net (accessed on 15 February 2023)).

Level	Type of Evidence
1A	Systematic review (with homogeneity) and meta-analysis of randomized controlled trials (RCTs)
1B	Individual RCT (with narrow confidence intervals)
1C	All or none study
2A	Systematic review (with homogeneity) of cohort studies
2B	Individual Cohort study (including low-quality RCT, e.g., <80% follow-up)
2C	“Outcomes” research; Ecological studies
3A	Systematic review (with homogeneity) of case-control studies
3B	Individual Case-control study
4	Case series (and poor-quality cohort and case-control study
5	Expert opinion without explicit critical appraisal or based on physiology bench research or “first principles”.

**Table 2 jcm-12-03087-t002:** GRADE Certainty Ratings.

Certainty	What It Means
Very low	The true effect is probably markedly different from the estimated effect
Low	The true effect might be markedly different from the estimated effect
Moderate	The authors believe that the true effect is probably close to the estimated effect
High	The authors have a lot of confidence that the true effect is similar to the estimated effect

**Table 3 jcm-12-03087-t003:** Evidence for several tinnitus treatments and recommendations of the various guidelines (listed in alphabetical order). As a source of evidence, we listed the most recent meta-analyses (focusing predominantly on Cochrane Meta-analyses when possible).

Intervention	Source of Evidence	Number of Study Participants	Efficacy (Immediate)	Efficacy (Long-Term)	Potential Harm	US (2014) [19]	Swiss (2019) [20]	European (2019) [21]	NICE (2020) [22]	German (2021) [23]	Japanese (2019) [24]
Anticonvulsants	Cochrane (Hoekstra 2011) [25]	453	Insufficient evidence	Not reported	Side effects reported in 18% of participants	clinicians should not routinely recommend anticonvulsants for a primary indication of treating persistent, bothersome tinnitus (recommendation against)	No pharmacological treatment	Weak recommendation against pharmacological treatment	Not mentioned	Strong recommendation against pharmacological treatment	Pharmacotherapy is not recommended, given its low level of evidence and side effects
Antidepressants	Cochrane (Baldo 2012) [26]	610	Insufficient evidence	Not reported	Side effects common	clinicians should not routinely recommend antidepressants for a primary indication of treating persistent, bothersome tinnitus (recommendation against)	No pharmacological treatment	Weak recommendation against pharmacological treatment	Not mentioned	Strong recommendation against pharmacological treatment	Recommendation with low evidence in case of coexisting depression or anxiety disorder
Auditory Training	Systematic review (Hoare 2010) [27]	269	Available evidence of insufficient quality to make a conclusion about efficacy	Not reported	Not reported	Not mentioned	Not mentioned	Not mentioned	Not mentioned	Recommendation for auditory training	Not mentioned
Betahistine	Cochrane (Wegner 2018) [28]	303	No significant effects on tinnitus loudness or distress	Not reported	Side effects on placebo level	Not mentioned	No pharmacological treatment	Weak recommendation against pharmacological treatment	Do not offer betahistine to treat tinnitus	Strong recommendation against pharmacological treatment	Pharmacotherapy is not recommended, given its low level of evidence and side effects
Cochlear Implant	Meta-Analysis (Oh 2022) [29]	674	Tinnitus score SMD: −1.32	Not reported	not reported	Not mentioned	Not mentioned	No recommendation for cochlear implants	Not mentioned	Strong recommendation for cochlear implants in patients with tinnitus and severe hearing loss/deafness	Recommendation with low evidence in patients who also have profound hearing loss
Cognitive behavioural therapy	Cochrane (Fuller 2020) [17]	2733	Tinnitus severity SMD: −0.56 THI: −10.91	No evidence due to a lack of data	Adverse effects are rare	Clinicians should recommend CBT to patients with persistent, bothersome tinnitus. (recommendation)	Efficacy clearly proven	Strong recommendation for cognitive behavioural therapy	if tinnitus is still causing an impact on emotional and social wellbeing and daily activities, consider a stepped approach: Digital tinnitus-related cognitive behavioural therapy (CBT) group-based tinnitus-related psychological interventions, including mindfulness-based cognitive therapy, acceptance and commitment therapy or CBT individual tinnitus-related CBT	Strong recommendation for cognitive behavioural therapy	Strong recommendation for cognitive behavioural therapy
Dexamethasone (intratympanic)	Meta-analysis (Chung 2022) [30]	220	no significant effect compared with the placebo	no significant effect compared with the placebo	Complications such as hearing loss, eardrum perforation, and middle ear inflammation are rare	clinicians should not routinely recommend intratympanic medications for a primary indication of treating persistent, bothersome tinnitus (recommendation against)	No pharmacological treatment	Weak recommendation against pharmacological treatment	Not mentioned	Strong recommendation against pharmacological treatment	Pharmacotherapy is not recommended, given its low level of evidence and side effects
Ginkgo biloba	Cochrane (Sereda 2022) [31]	1915	little to no effect at three to six months compared to a placebo, but the evidence is very uncertain	little to no effect at three to six months compared to a placebo, but the evidence is very uncertain	Incidence of side effects low	Clinicians should not recommend Ginkgo biloba for treating patients with persistent, bothersome tinnitus (recommendation against)	No pharmacological treatment	Weak recommendation against pharmacological treatment	Not mentioned	Strong recommendation against pharmacological treatment	Pharmacotherapy is not recommended, given its low level of evidence and side effects
Hearing Aid	Cochrane (Sereda 2018) [32]	590	No significant effects on tinnitus loudness or distress	No data	Not reported	Clinicians should recommend a hearing aid evaluation for patients with hearing loss and persistent, bothersome tinnitus (recommendation)	Patients with a tinnitus pitch below 6 kHz seem to benefit	Weak recommendation for hearing aids	Offer amplification devices to people with tinnitus who have hearing loss that affects their ability to communicate Consider amplification devices for people with tinnitus who have hearing loss but do not have difficulties communicating Do not offer amplification devices to people with tinnitus but no hearing loss	Recommendation for hearing aids in case of hearing loss	Strong recommendation for tinnitus that is accompanied by hearing loss
Hyperbaric Oxygen	Cochrane (Bennett 2012) [33]	392	no significant improvements in tinnitus for chronic tinnitus	no significant improvements in tinnitus for chronic tinnitus	Not reported	Not mentioned	No proof of efficacy	Not mentioned	Not mentioned	Not mentioned	Not mentioned
Sound Therapy	Cochrane (Sereda 2018) [32]	590	No significant effects on tinnitus loudness or distress	No data	Not reported	Clinicians may recommend sound therapy to patients with persistent, bothersome tinnitus. (option)	Only limited data from controlled trials	No recommendation	Recommendation for research	Recommendation against sound generators, Recommendation against specific sound therapies	No recommendation for sound generators
Tinnitus Retraining Therapy	Meta-analysis (Han 2021 [34])	1345	Significantly increased treatment response	Significantly increased treatment response	Not reported	Not mentioned	lack of high-quality trials	No recommendation	Not mentioned	Can be considered for long-term treatment	Recommendation with low evidence
Transcranial direct current stimulation	Meta-analysis (Martins 2022) [35]	1031	Loudness SMD: −0.35 Distress SMD: −0.50	Not reported	Not reported	Not mentioned	Not mentioned	No recommendation for transcranial electrical stimulation	Recommendation for research	Recommendation against transcranial electrical stimulation	Not mentioned
Transcranial magnetic stimulation	Meta-Analysis (Lefebvre-Demers 2021 [36])	945	Tinnitus severity SMD: −0.45	Tinnitus severity SMD: −0.42	Not reported	Clinicians should not recommend TMS for the routine treatment of patients with persistent, bothersome tinnitus (recommendation against)	Not mentioned	recommendation against transcranial magnetic stimulation	Recommendation for research	Recommendation against transcranial magnetic stimulation	Recommendation against low evidence
Zinc	Cochrane (Person 2016 [37])	209	no evidence for improvement of tinnitus severity by oral zinc supplementation	no evidence for improvement of tinnitus severity by oral zinc supplementation	Not reported	Clinicians should not recommend Zinc for treating patients with persistent, bothersome tinnitus (recommendation against)	No pharmacological treatment	Weak recommendation against pharmacological treatment	Not mentioned	Strong recommendation against pharmacological treatment	Pharmacotherapy is not recommended, given its low level of evidence and side effects

## Data Availability

Not applicable.

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
