# Peer review of "Tinnitus Guidelines and Their Evidence Base"

_jcm, 2023, doi:10.3390/jcm12093087_

Round 1

Reviewer 1 Report (New Reviewer)

Overall a very well-written manuscript giving a meaningful overview of existing guidelines for tinnitus management and how future guidelines could be established. I think that it would be valuable if this piece was published, but I have a few comments for the authors and the editor to consider.

General comment: I’m missing the aspect of funding bias. A lot of studies reporting tinnitus interventions are funded by the company trying to sell the drug/hearing aid/sound generator/equipment, and we know from other fields of research that industry funded studies are far more likely to report results that are positive for the funder.

- Line 213-235: One aspect missing is that different interventions are not equally easy to test empirically. It is easy to conduct a randomized double blind controlled study if you are testing a pharmacological intervention as the control group can receive a placebo pill. The same does not apply to hearing aids, as individuals receiving a hearing aid that doesn’t amplify sounds would notice straight away and thereby know that they are in the control group. In addition, since early intervention is of great benefit for successful hearing aid intervention, and the negative effects of an unaddressed hearing loss are indisputable (e.g. Livingston et al., 2020), it is also harder to get ethical approval to let some participants receive hearing aid amplification but not others. In other words, the ability to achieve highest level of evidence also varies across interventions. There are lots of studies (along with global clinical experience) indicating hearing aid amplification to be beneficial for individuals with tinnitus, but when applying very strict criteria for RCTs one ends up with only three studies (with few participants each) in the quantitative meta-analysis (Serada et al., 2018) which just isn’t enough data to base a proper interpretation. As you mentioned so elegantly in your abstract, absence of evidence is not the same as evidence of absence. I think that sentence fits well into the referenced segment (line 213-235).  

Furthermore, as pharmacological interventions typically are associated with more severe negative side effects than hearing aid amplification (which typically has positive side effects, see comment below), maybe the strength of evidence needed for an intervention to be recommended should be weighted depending on their potential risks (i.e. we need clearer evidence of the treatment effect for an intervention that might also harm our patients, compared to an intervention with no negative side effects or even positive side effects). GRADE works very well for pharmacological interventions, but not as well for all other type of treatments. There are examples of similar difficulties in other fields, where a solution has been to not only use GRADE when assessing level of evidence. Diet and nutrition studies have similar problems, where drugs are easily tested with RCTs but eating patterns are more difficult. For that context NutriGRADE (Schwingshackl et al. 2016) and HEALM (Katz et al., 2019) has been used when grading evidence for different treatments.
Maybe similar strategies could be adopted when establishing tinnitus guidelines? When considering this, it should also be mentioned that healthcare outcomes from observational studies typically are very similar to results from RCTs both in direction of effect and size of effect (Benson & Hartz, 2000; Anglemyer et al., 2014; Vandenbroucke, 2009).

- Table 3: this is a great table, but I’m missing a column with potential positive side effects. As the focus of your paper is clinical guidelines, a holistic approach is reasonable. For instance, hearing aids are protective against cognitive decline (Yeo et al., 2022), and should thus be recommended even if the effect on tinnitus distress would be moderate.

- Line 213-235: Here it could be mentioned that a recent meta-analysis indicated that objectively verified hearing aid amplification is associated with greater reductions in tinnitus distress compared to unverified amplification, especially in the long run (Waechter & Jönsson, 2022). This indicates that tinnitus care could be significantly improved with very small means, which would be valuable to convey to the reader.

- Line 182-183: Can you really say that the study reported by Henry et al. (2017) is a recent study (6 years old), and can you really say that it demonstrated the efficacy of the counselling they delivered to the participants when every participant received the same counselling protocol? How can we be sure that any effects are related to the counselling protocol and not the Comparing Extended-Wear Hearing Aids, Conventional Hearing Aids, and Combination Instruments? Maybe it would be more relevant to mention the study by Searchfield and colleagues (2010) who compared the effect of counselling with versus without hearing aid amplification on tinnitus distress. They found that tinnitus distress decreased in both groups but the magnitude was clearly greater when counselling was combined with (REM verified) hearing aid amplification.

- Line 209-212: Maybe one could add that this may indicate that VR-based interventions could be offered to tinnitus patients who feel like CBT isn’t for them? If I’ve been correctly informed, this is something you’ve encountered quite a lot in the UNITI-project?

- Line 236-240: It would be helpful to the reader if you’d communicate that while there is some evidence for TRT against tinnitus distress, the sound generator part of the typical TRT protocol does not seem to be necessary to achieve the effect (Scherer et al., 2019)

- Line 241-248: Maybe mention that melatonin seems to be helpful especially in tinnitus patients with sleeping difficulties (Rosenberg et al., 1998; Hurtuk et al., 2011; Megwalu et al., 2006)?

- Line 319-381: How come you don’t mention the Japanese guidelines (Ogawa et al., 2019)?

- Line 406-437: This segment is very interesting, please keep this!

- Line 552-564: I agree that the delay of tinnitus guidelines is a real problem, but are tinnitus studies really published frequently enough to motivate a weekly guideline update? The situation with covid was clearly different, and the comparison seems a bit far-fetched. Maybe figure 3 could be put in an appendix? I do also understand that you want to communicate the need for a decision support system, and I believe many clinicians are looking forward to when the UNITI DSS will be released, but I think you could convey this without referencing the covid-pandemic.

Author Response

Overall a very well-written manuscript giving a meaningful overview of existing guidelines for tinnitus management and how future guidelines could be established. I think that it would be valuable if this piece was published, but I have a few comments for the authors and the editor to consider.

Answer: Thank you for your encouraging review; we are grateful for your comments.

General comment: I’m missing the aspect of funding bias. A lot of studies reporting tinnitus interventions are funded by the company trying to sell the drug/hearing aid/sound generator/equipment, and we know from other fields of research that industry funded studies are far more likely to report results that are positive for the funder.

Answer: according to the reviewer’s suggestion we added “funding bias” explicitly in the introduction as a risk of bias:

Risks of bias include sample selection bias (the sample is not representative for the patient population, publication bias (negative results are more likely not to be published) or funding bias (influencing the study design by a funder to achieve a particular outcome). 

- Line 213-235: One aspect missing is that different interventions are not equally easy to test empirically. It is easy to conduct a randomized double blind controlled study if you are testing a pharmacological intervention as the control group can receive a placebo pill. The same does not apply to hearing aids, as individuals receiving a hearing aid that doesn’t amplify sounds would notice straight away and thereby know that they are in the control group. In addition, since early intervention is of great benefit for successful hearing aid intervention, and the negative effects of an unaddressed hearing loss are indisputable (e.g. Livingston et al., 2020), it is also harder to get ethical approval to let some participants receive hearing aid amplification but not others. In other words, the ability to achieve highest level of evidence also varies across interventions. There are lots of studies (along with global clinical experience) indicating hearing aid amplification to be beneficial for individuals with tinnitus, but when applying very strict criteria for RCTs one ends up with only three studies (with few participants each) in the quantitative meta-analysis (Serada et al., 2018) which just isn’t enough data to base a proper interpretation. As you mentioned so elegantly in your abstract, absence of evidence is not the same as evidence of absence. I think that sentence fits well into the referenced segment (line 213-235).  

Furthermore, as pharmacological interventions typically are associated with more severe negative side effects than hearing aid amplification (which typically has positive side effects, see comment below), maybe the strength of evidence needed for an intervention to be recommended should be weighted depending on their potential risks (i.e. we need clearer evidence of the treatment effect for an intervention that might also harm our patients, compared to an intervention with no negative side effects or even positive side effects). GRADE works very well for pharmacological interventions, but not as well for all other type of treatments. There are examples of similar difficulties in other fields, where a solution has been to not only use GRADE when assessing level of evidence. Diet and nutrition studies have similar problems, where drugs are easily tested with RCTs but eating patterns are more difficult. For that context NutriGRADE (Schwingshackl et al. 2016) and HEALM (Katz et al., 2019) has been used when grading evidence for different treatments. Maybe similar strategies could be adopted when establishing tinnitus guidelines? When considering this, it should also be mentioned that healthcare outcomes from observational studies typically are very similar to results from RCTs both in direction of effect and size of effect (Benson & Hartz, 2000; Anglemyer et al., 2014; Vandenbroucke, 2009).

Answer:

We agree with the author and discuss these aspects in an additional paragraph in the introduction:

A fair and balanced comparative evaluation of efficacy and safety of various types of therapeutic interventions is challenging as the criteria for grading the evidence level are not equally applicable to different interventions. This can be best illustrated by the re-quirement of a control condition with double blinded group allocation of study partici-pants. This may be feasible for a pharmacological intervention, but impossible for testing a hearing device, counselling or cognitive behavioural therapy. When testing these latter interventions, compromises in methodological quality must be accepted. However, com-parison of a certain intervention with a waiting list control group (ref) may favour this in-tervention, when results are compared to an intervention that is tested in placebo-controlled trial with effective blinding. Similar considerations are true for the assess-ments of safety of therapeutic interventions. Whereas a thorough safety assessment is mandatory in pharmacological trials, side effects are only incompletely documented in trials that evaluate psychotherapeutic interventions (ref).

- Table 3: this is a great table, but I’m missing a column with potential positive side effects. As the focus of your paper is clinical guidelines, a holistic approach is reasonable. For instance, hearing aids are protective against cognitive decline (Yeo et al., 2022), and should thus be recommended even if the effect on tinnitus distress would be moderate.

Answer: We agree with the reviewer that the consideration of beneficial side effects is an important clinical aspect. Adding a further column in table 3 would have made the already quite complex table (see also comment of reviewer 2) more confusing. We therefore added additional sentences in the introduction and in the conclusion, in which we stress the relevance of co-morbidities and the need to consider all relevant aspects in a holistic way for individual treatment decisions:

Moreover, the extent of the tinnitus burden is strongly influenced by co-morbidities such as hearing loss, depression or anxiety (ref). Interventions that act primarily on these co-morbidities (e.g. hearing aids or antidepressants) may have a beneficial effect for the tinnitus patient, but the specific effect on tinnitus might be difficult to disentangle from the beneficial effect on the co-morbidity. 

For the individual case, chances of improvement, risks of treatment, alternative options, overall health state, psychosocial situation and the patient’s values and desires have to be weighted in a holistic approach to come to a reasonable decision.

- Line 213-235: Here it could be mentioned that a recent meta-analysis indicated that objectively verified hearing aid amplification is associated with greater reductions in tinnitus distress compared to unverified amplification, especially in the long run (Waechter & Jönsson, 2022). This indicates that tinnitus care could be significantly improved with very small means, which would be valuable to convey to the reader.

Answer: we mention this finding by adding the following sentence:

A recent metaanalysis has shown, that the results strongly depend on the procedure, how hearing aid fitting is performed (Waechter & Jönsson, 2022).

- Line 182-183: Can you really say that the study reported by Henry et al. (2017) is a recent study (6 years old), and can you really say that it demonstrated the efficacy of the counselling they delivered to the participants when every participant received the same counselling protocol? How can we be sure that any effects are related to the counselling protocol and not the Comparing Extended-Wear Hearing Aids, Conventional Hearing Aids, and Combination Instruments? Maybe it would be more relevant to mention the study by Searchfield and colleagues (2010) who compared the effect of counselling with versus without hearing aid amplification on tinnitus distress. They found that tinnitus distress decreased in both groups but the magnitude was clearly greater when counselling was combined with (REM verified) hearing aid amplification.

Answer: We agree with the reviewer, changed the sentence and replaced the reference:

Educational counselling was used as an active control condition in several studies, where it had a beneficial effect (e.g. Searchfield 2010, Piromchai 2022).

- Line 209-212: Maybe one could add that this may indicate that VR-based interventions could be offered to tinnitus patients who feel like CBT isn’t for them? If I’ve been correctly informed, this is something you’ve encountered quite a lot in the UNITI-project?

Answer: we agree with the reviewer and added the following sentence:

It should be noted that only a subgroup of patients are willing to undergo CBT and that the availability of tinnitus specific CBT is limited. Innovative forms of CBT such as smartphone App based CBT or virtual reality based CBT (Malinvaud et al. 2016) might enhance its acceptance and frequency of use.

- Line 236-240: It would be helpful to the reader if you’d communicate that while there is some evidence for TRT against tinnitus distress, the sound generator part of the typical TRT protocol does not seem to be necessary to achieve the effect (Scherer et al., 2019)

Answer: we followed the reviewer's suggestion and added the following sentence:

Moreover, there is an ongoing debate about the contribution of sound therapy to TRT ef-fects. According to a recent multicentric study TRT with and without sound therapy did not differ in their effects (Scherer Foarmby 2019)

- Line 241-248: Maybe mention that melatonin seems to be helpful especially in tinnitus patients with sleeping difficulties (Rosenberg et al., 1998; Hurtuk et al., 2011; Megwalu et al., 2006)?

Answer: according to the reviewer's suggestion we mentioned the evidence of pharmacotherapy for tinnitus co-morbidities.

There is some limited evidence for pharmacological treatment of comorbid conditions: Antidepressants have shown beneficial effects for comorbid depressive symptoms (Zöger et al. 2006), melatonin improved sleep in tinnitus patients in controlled studies (Rosenberg 1998, Megwalu 2006,Hurtuk 2011), and Ginkgo biloba was shown beneficial in elderly tinnitus patients with dementia (42).

- Line 319-381: How come you don’t mention the Japanese guidelines (Ogawa et al., 2019)?

Answer: We would like to thank the reviewer for pointing out this omission. We now added a paragraph about the Japanese guidelines and also added a further column in table 3

Japanese Guideline

The Japanese Clinical Practice Guideline for Diagnosis and Treatment of Chronic Tinnitus was developed by subcommittee members, edited by the Japan Audiological So-ciety, authorized by the Oto-Rhino-Laryngological Society of Japan and published in 2019 (Ogawa et al. 2020). Recommendations are graded in strong recommendation (1), recom-mendation (2) and no recommendation (none). The evidence level of each recommenda-tion is graded in 4 categories (A,B,C,D), which are added to each recommendation.

- Line 406-437: This segment is very interesting, please keep this!

Answer: we are happy to follow this recommendation

- Line 552-564: I agree that the delay of tinnitus guidelines is a real problem, but are tinnitus studies really published frequently enough to motivate a weekly guideline update? The situation with covid was clearly different, and the comparison seems a bit far-fetched. Maybe figure 3 could be put in an appendix? I do also understand that you want to communicate the need for a decision support system, and I believe many clinicians are looking forward to when the UNITI DSS will be released, but I think you could convey this without referencing the covid-pandemic.

Answer: We are aware that the development of new knowledge in the tinnitus is not occurring at a speed that would require weekly guideline updates. We used the example of Covid only to illustrate, that guideline actualization in short cycles is possible. We added the following sentence:

We are aware that the development of new knowledge in the tinnitus is not occurring at a speed that would require weekly guideline updates, but yearly actualization cycles would be highly desirable.

Reviewer 2 Report (New Reviewer)

This is the most comprehensive review of tinnitus therapies I have seen, and I think it will be quite useful.  I have a number of general comments for enhancing the utility and conciseness of the paper.

The title should be shortened to include just the words that precede the colon.

The writing is simply too 'gassy', entailing many awkward and unnecessary clauses.  The text could be reduced by 10-15% by improved conciseness.  The use of commas is nearly random, with too many in some places and not enough in others.  This harms the flow and readability.  There are a lot of run-on sentences that are hard to parse.  Generally, any sentence that winds on for more than 3 lines (in the format I was given) should be shortened.  One effect of the over-use of commas is to create unreadable sentences and non-sentences that lack an actual subject and a verb.  Paragraph structure, likewise, is nearly random.  Each paragraph should have a clear topic sentence and deal completely with an idea.  No paragraph should begin with a work like 'This' or 'Thus' or any other vague reference to the preceding paragraph.  Where this is done, it is generally a signal that there should be no new paragraph.

The tables are useful, but contain some words or metrics that should be defined.  Box 1 needs to be condensed for space, and if all the therapies mentioned were tested by RCTs, where are the references?  Table 3 is helpful and needed, but the format is problematic.  I am torn as to how to fix this.  It is just very difficult to follow.  No table should contain hyphenated words.

Tinnitus is very heterogeneous, at the authors recognize, but this merits more development.  The Intro section should include a section that attempts to break tinnitus down into the currently best supported categories by etiology and manifestation.  I don't think it does any good to talk about therapies without regard to different causes and forms.  Tinnitus-related metrics should also be presented and explained, along with an explanation of how effect size is measured.

It might be helpful to talk a bit in the Intro about tinnitus research in animals and its limitations.

Author Response

This is the most comprehensive review of tinnitus therapies I have seen, and I think it will be quite useful.  I have a number of general comments for enhancing the utility and conciseness of the paper.

Answer: We want to thank the reviewer for this positive evaluation.

The title should be shortened to include just the words that precede the colon.

Answer: done

The writing is simply too 'gassy', entailing many awkward and unnecessary clauses.  The text could be reduced by 10-15% by improved conciseness.  The use of commas is nearly random, with too many in some places and not enough in others.  This harms the flow and readability.  There are a lot of run-on sentences that are hard to parse.  Generally, any sentence that winds on for more than 3 lines (in the format I was given) should be shortened.  One effect of the over-use of commas is to create unreadable sentences and non-sentences that lack an actual subject and a verb.  Paragraph structure, likewise, is nearly random.  Each paragraph should have a clear topic sentence and deal completely with an idea.  No paragraph should begin with a work like 'This' or 'Thus' or any other vague reference to the preceding paragraph.  Where this is done, it is generally a signal that there should be no new paragraph.

Answer: We edited the whole article according to the reviewer's suggestions.

The tables are useful, but contain some words or metrics that should be defined.  Box 1 needs to be condensed for space, and if all the therapies mentioned were tested by RCTs, where are the references?  Table 3 is helpful and needed, but the format is problematic.  I am torn as to how to fix this.  It is just very difficult to follow.  No table should contain hyphenated words.

Answer: According to the reviewer's suggestion, we removed hyphenated words from the tables and boxes. Concerning the layout of boxes and tables, we leave it to the publisher, to arrange them clearly.

Tinnitus is very heterogeneous, at the authors recognize, but this merits more development.  The Intro section should include a section that attempts to break tinnitus down into the currently best supported categories by etiology and manifestation.  I don't think it does any good to talk about therapies without regard to different causes and forms.  Tinnitus-related metrics should also be presented and explained, along with an explanation of how effect size is measured.

Answer: We completely agree with the reviewer that the heterogeneity of tinnitus is a highly important aspect and has to be addressed for improving the clinical management of tinnitus patients. This is discussed explicitly in the discussion:

The methodological gold standard – studying a therapeutic intervention in many mul-ti-centric randomized controlled trials and performing a meta-analysis of their results – aims to focus on the essential aspect of the pathological condition and the therapeutic in-tervention by removing all patient- and therapist-specific aspects by averaging across many patients and treatment settings. The more homogeneous the pathophysiology of a given disorder and the more homogeneous the intervention, the more appropriate is this approach. However, in case of a heterogeneous condition such as tinnitus this averaging approach has major limitations. Therefore, it is of utmost importance to identify criteria for mean-ingful subtypes of tinnitus, as this would allow the investigation of specific interventions for these specific subtypes.

In the introduction we had already stated that tinnitus heterogeneity and its subjective nature are major challenges for the development of evidence based treatment recommendations

With the advent of EBM, clinical researchers aimed at evaluating the effectiveness of vari-ous treatments by conducting clinical trials. This is far from trivial in the tinnitus field for several reasons (8-10). First, tinnitus is a heterogeneous condition. Second, it is mostly purely subjective, which makes outcome measurement challenging, and third, suffering from tinnitus has many facets, which vary from patient to patient.

According to the reviewer's suggestion we now further expanded this section:

All these aspects have been addressed in the last years. Research has focused on the identification of clinically relevant distinct subtypes of tinnitus. This was only partly successful, as the heterogeneity is better described by dimensional variability of tinnitus characteristics, than by distinct categories (e.g. more or less somatic involvement instead of a category "somatic tinnitus"). With respect to the many facets of tinnitus burden, re-search aimed to identify consensus-based core outcome domains. For outcome measurement, various tools have been developed and tested for psychometric adequacy.  These efforts revealed, that both tinnitus questionnaires and visual analogue- and numeric rating scales represent reliable and valid tools for the quantification of tinnitus impact, whereas psychometric measurements of tinnitus loudness are not very helpful for this purpose.

It might be helpful to talk a bit in the Intro about tinnitus research in animals and its limitations.

Answer: We agree that the lack of highly effective tinnitus treatments may be related to the incomplete understanding of the pathophysiology of tinnitus and the limited predictive value of currently available animal models. However, we would like to refrain from discussing the current state of animal research in tinnitus in detail, as this is beyond the scope of our article.

This manuscript is a resubmission of an earlier submission. The following is a list of the peer review reports and author responses from that submission.

Round 1

Reviewer 1 Report

The study focuses on an important and ongoing aspect of tinnitus research, namely treatment of tinnitus. Various interventions are being made against tinnitus, but none so far is fully satisfactory.

The authors have presented the current state of knowledge in an objective and balanced way.

I have only one comment. How the authors assess the chance that a general consensus will be reached on how to treat tinnitus. What conditions would have to be met? Would the Delphi method be useful in achieving it?

Author Response

We want to thank the reviewer for the positive evaluation of our paper and are happy to answer the reviewer’s comment:

The study focuses on an important and ongoing aspect of tinnitus research, namely treatment of tinnitus. Various interventions are being made against tinnitus, but none so far is fully satisfactory.

The authors have presented the current state of knowledge in an objective and balanced way.

I have only one comment. How the authors assess the chance that a general consensus will be reached on how to treat tinnitus. What conditions would have to be met? Would the Delphi method be useful in achieving it?

Answer:  Guidelines should provide an orientation about how tinnitus should be treated.

Concerning the potential use of the Delphi method for reaching a consensus how tinnitus should be treated, there are two possible applications:

  1. Expert consensus as the basis for a clinical recommendation

As mentioned the recommendations of the guidelines should be based on current evidence from clinical research. The Delphi method is a method for reaching consensus among experts. However, according to guideline methodology, expert consensus is considered as the lowest level evidence and should only be used if data from clinical trials are not available.

  1. Expert consensus for translating clinical evidence into clinical recommendations

In principle the Delphi method could be used by the guideline committees for translating clinical evidence into clinical recommendations, as the Delphi method is a method to create consensus statements and as the translation of clinical evidence into clinical recommendations requires consensus among the guideline committee.

Reviewer 2 Report

The authors aim in this perspectives article to evaluate the current international procedures for developing guidelines for tinnitus treatments. They show their limitations and provide several options to optimize these procedures in timing and research related aspects.

General comments

The article is a mixture of a general review of guideline methods and a perspective article with a focus on optimizing these methods and allowing new treatment options to be included faster / easier into these guidelines. Nevertheless, the article is written with a strong European/US bias. Not a single guideline from Asia, Africa, South America or Oceania is mentioned, even though the majority of all possible tinnitus patients may live there. This may be due to the national backgrounds of the authors, but to have a real overview of the worldwide tinnitus approaches, at least some of the guidelines (or their absence) of some major countries there should be taken into account (especially Table 3 and related text sections). This has then also to be included in the Discussion.

To accommodate the review character of the article, there should be at least a kind of overview, how many studies are included in each aspect of the tinnitus treatment options, e.g. shown in Box 1. And to accommodate the perspective character of the manuscript, I suggest a kind of four or five category analog scale for the above mentioned interventions given in Box 1 that are not already included in the guidelines of Table 3. This would have to be performed by the authors as specialists of the field and may provide additional arguments, what interventions might be most promising for future research. This evaluation of interventions has to be also included in the Methods section.

Overall, I suggest a major revision of the manuscript.

Specific comments

L37    change the dash to a hyphen and remove the space

L44    you could introduce the abbreviation EBM here.

L85    “clinical guidelines, based on a consensus”. Add some references here or refer to the overview in the (revised) Table 3.

L119   make this heading-like

L189   - State, why you chose the guidelines, why are there no asian, african or south american guidelines included?

- state your perspective question (again) here.

Table 3:´ it would be helpful to make only general remarks, like (weak / strong) recommendation against / for treatment

Citation for the different guidelines? Or maybe cite them above and refer to the citations in the table caption.

Box 1: add the number of papers investigating the different interventions

L204   A sentence like "Not all interventions of box 1 are revised in the guidelines." would be helpful here

L207   Would you be able to judg / estimate the power of each intervention? I think, it would help the reader a lot to get a kind of efficacy value. Here you could, e.g., use a 4 or 5 step analog scale. If you do so, plese also add it to the Methods section.

L272   There is a recent review on Ginkgo biloba and Tinnitus that should be mentioned here. https://doi.org/10.1016/j.mcn.2021.103669

L338   add “(UK)”

L340   add the year

L397   It would be helpful to have a very short summary at the beginning of the Discussion to help the reader to see the large picture.

L447   Here you could write something like “This is similar to the requirements in the USA..”

L457   after “clinical practice” you could also mention something like “...and payment by healthcare providers, (see also https://doi.org/10.3390/ijerph191610455 and DOI: 10.1097/AUD.0b013e31827d113a)”

L495   change “in” to “into”

L536   change “Corona pandemic” to “SARS 2 Covid 19 pandemic” (also L551)

L557   mention Figure 3 here

Author Response

We want to thank reviewer 2 for his constructive comments and are happy to provide a point by point answer:

The article is a mixture of a general review of guideline methods and a perspective article with a focus on optimizing these methods and allowing new treatment options to be included faster / easier into these guidelines. Nevertheless, the article is written with a strong European/US bias. Not a single guideline from Asia, Africa, South America or Oceania is mentioned, even though the majority of all possible tinnitus patients may live there. This may be due to the national backgrounds of the authors, but to have a real overview of the worldwide tinnitus approaches, at least some of the guidelines (or their absence) of some major countries there should be taken into account (especially Table 3 and related teact that we t sections). This has then also to be included in the Discussion.

Answer: Indeed we consider our article primarily as a perspective about how the application of evidence based medicine can be improved in the tinnitus field. For this purpose we summarized the current state of guidelines, in order to demonstrate its strengths and its weaknesses. The fact that the text only mentioned guidelines from Europe and the US is not related to the origin of the authors, but to the fact, that our literature research only identified guidelines from these countries. Nevertheless, we follow the suggestion of the reviewer and added this important aspect to the discussion (lines 516-520):

“Finally, it is remarkable, that we could only identify guidelines from European countries or the US. This means that for large parts of the world no guidelines are available. As tinnitus suffering and coping also involves cultural aspects, it is questionable to which extent recommendations from Europe or the US can be applied to patients in Asia, Africa, South America or Oceania.”

To accommodate the review character of the article, there should be at least a kind of overview, how many studies are included in each aspect of the tinnitus treatment options, e.g. shown in Box 1. And to accommodate the perspective character of the manuscript, I suggest a kind of four or five category analog scale for the above mentioned interventions given in Box 1 that are not already included in the guidelines of Table 3. This would have to be performed by the authors as specialists of the field and may provide additional arguments, what interventions might be most promising for future research. This evaluation of interventions has to be also included in the Methods section.

Answer: We understand the proposal of the reviewer and agree that such additional information might be valuable. However, we rather abstain from such an evaluation of the available treatment methods, as this goes far beyond the scope of this perspective. We added a sentence in the introduction that the goal of this perspective is the discussion of the strengths and the weaknesses of the current adaption of evidence based medicine in tinnitus and that the summary of current evidence in this perspective serves mainly as basis for discussion of the various guideline recommendations. (lines 190-194):

“In this perspective paper we aim to discuss strengths and weaknesses of the current adaption of evidence based medicine in tinnitus. For this purpose we will summarize the currently existing guidelines based on a literature search and contrast them with each other and with the currently available evidence (see particularly table 3).”

Specific comments

L37    change the dash to a hyphen and remove the space

Answer: done

L44    you could introduce the abbreviation EBM here.

Answer: done

L85    “clinical guidelines, based on a consensus”. Add some references here or refer to the overview in the (revised) Table 3.

Answer: reference added

L119   make this heading-like

Answer: done

L189   - State, why you chose the guidelines, why are there no asian, african or south american guidelines included?

Answer: we provided the explanation (line 191,192):

“For this purpose we will summarize the currently existing guideline based on a literature search.”

- state your perspective question (again) here.

Answer: we added an additional sentence to the introduction (line 190,191):

“In this perspective paper we aim to discuss strengths and weaknesses of the current adaption of evidence based medicine in tinnitus.”

Table 3:´ it would be helpful to make only general remarks, like (weak / strong) recommendation against / for treatment

Answer: By purpose we reported the exact wording of the guideline recommendations in order to illustrate the differences in the recommendation

Citation for the different guidelines? Or maybe cite them above and refer to the citations in the table caption.

Answer: all guidelines are now also cited in the table

Box 1: add the number of papers investigating the different interventions

Answer: As mentioned above, such a comprehensive overview about all clinical trials focussing on tinnitus treatment goes beyond the scope of this paper. According to the GRADE classification we mentioned the most recent meta-analysis (Cochrane, if available) to indicate the best currently available evidence for each intervention.

L204   A sentence like "Not all interventions of box 1 are revised in the guidelines." would be helpful here

Answer: We want to thank for this suggestion and added the following sentence (line 207/208):

“Notably not all these investigated interventions are revised in the guidelines.”

L207   Would you be able to judge / estimate the power of each intervention? I think, it would help the reader a lot to get a kind of efficacy value. Here you could, e.g., use a 4 or 5 step analog scale. If you do so, please also add it to the Methods section.

Answer: In order to provide an estimate of the immediate and long-term efficacy of the various therapeutic interventions, we mentioned effect sizes in table 3

L272   There is a recent review on Ginkgo biloba and Tinnitus that should be mentioned here. https://doi.org/10.1016/j.mcn.2021.103669

Answer: we want to thank the reviewer to direct our attention to this recent review on pharmacologic treatments in preclinical tinnitus models. As we focus in our perspective on clinical guidelines, we limited the literature research to clinical studies.

L338   add “(UK)”

Answer: done

L340   add the year

Answer: done

L397   It would be helpful to have a very short summary at the beginning of the Discussion to help the reader to see the large picture.

Answer: According to the reviewer’s suggestion we added the following sentences at the begin of the discussion (line 415-417):

“We will first discuss the translation of the currently available clinical evidence into the different guidelines. In the second part we will provide suggestions for further development..”

L447   Here you could write something like “This is similar to the requirements in the USA..”

Answer: Our point here is, that the threshold for recommendation of an innovative treatment in the guidelines is clearly higher as compared to the requirements for FDA approval in the USA. Obviously this was not clear in our manuscript. We added the following sentence for clarification (lines 470-472):

“Thus, the threshold for recommendation of an innovative treatment in the guidelines is clearly higher as compared to the requirements for FDA approval in the USA, where a non-inferiority trial is sufficient.”

L457   after “clinical practice” you could also mention something like “...and payment by healthcare providers, (see also https://doi.org/10.3390/ijerph191610455 and DOI: 10.1097/AUD.0b013e31827d113a)”

Answer: we want to thank the reviewer for the suggestion to mention here the socioeconomic aspect and added the following sentence and the suggested references (lines 483-485,)

“From an economic perspective, an earlier availability of effective innovative treatments has the potential to reduce the enormous direct and indirect socio-economic costs of tinnitus.”

L495   change “in” to “into”

Answer: done

L536   change “Corona pandemic” to “SARS 2 Covid 19 pandemic” (also L551)

Answer: done

L557   mention Figure 3 here

Answer: done

Reviewer 3 Report

General comments:

Good research idea. English language is good, however, the article showed redundancy in different sections especially in the introduction and methodology sections.

Introduction:

·       The authors comprehensively discuss the evidence based medicine methodology, however it lacks the focus on the main topic of this article which is tinnitus.

·       The authors should write about tinnitus and its causes and general lines of treatment. It is better to shift Table 3 to be present in the introduction section.  

·       Pages 1-2, lines 32 - 98 are very redundant. They must be more concise.

Methodology:

·       The method utilized by the authors for selecting articles included in this review is not clear at all. It should include the number of articles and it must include an inclusion and exclusion criteria

·       Also, the description of different articles and the methodology used in each one is not present at all. The tools used in these articles to assess the outcome of their approach used for tinnitus management is also not present.

·       Pages 4- 5, lines 143-189: too much information that need abstracting.

Results: poor

·       Box 1 needs to be rearranged again. The methodologists used in RCTs mentioned in this box (such as physiotherapy and virtual reality based approaches) were not mentioned and the results sections at all.

·       The authors mentioned some approaches used in tinnitus management such as tinnitus counselling, cognitive behavioral therapy, mindfulness and tinnitus… etc. Each one of these approaches should discussed in relation to the number of studies, the indication, the methodology, the results, and the method of qualification the outcome.

Discussion: Need to be more systematized starting with revising the tinnitus management approach followed by discussion of different guidelines and their effect sizes, and recommendation to be used or not.

Consideration for future directions and conclusion are lengthy and need to be rewritten again and more comprehensive shorter statements.

Author Response

General comments:

Good research idea. English language is good, however, the article showed redundancy in different sections especially in the introduction and methodology sections.

Introduction:

The authors comprehensively discuss the evidence based medicine methodology, however it lacks the focus on the main topic of this article which is tinnitus.

Answer: The main focus of this perspective paper is the discussion of the current adaption of evidence based medicine in tinnitus. For this reason, we first summarize the current standards of clinical guideline development and then discuss them with respect to tinnitus. We added an additional sentence in the introduction to clarify the scope of this perspective paper   (lines 190-194):

“In this perspective paper we aim to discuss strengths and weaknesses of the current adaption of evidence based medicine in tinnitus. For this purpose we will summarize the currently existing guidelines based on a literature search and contrast them with each other and with the currently available evidence (see particularly table 3).”

The authors should write about tinnitus and its causes and general lines of treatment. It is better to shift Table 3 to be present in the introduction section. 

Answer: As mentioned above the purpose of the perspective paper is not a review on causes and general lines of treatment, but a discussion about the adaptation of evidence based medicine in tinnitus. Therefore, we think that it is appropriate that the synopsis of current guidelines and the evidence for the various treatments is in the main part of the manuscript.

Pages 1-2, lines 32 - 98 are very redundant. They must be more concise.

Answer:  For the purpose of this perspective paper we consider a comprehensive introduction in evidence based medicine and guideline development as appropriate and would opt to keep this part of the introduction in its current form

The method utilized by the authors for selecting articles included in this review is not clear at all. It should include the number of articles and it must include an inclusion and exclusion criteria

Answer: we would like to stress again, that the scope of this perspective paper is not a systematic review of therapeutic interventions in tinnitus. We rather aimed to discuss the EBM approach in tinnitus. For this purpose we compared the recommendations in the various guidelines with each other and added the current evidence based on the most recent (Cochrane) meta-analysis as a reference. We chose meta-analyses, as they represent the highest evidence level in EBM and thus represent the evidence on which clinical guideline recommendations should be based. We clarified this aspect in the heading of table 3 (line 195,196)

“As source of evidence we listed the most recent meta-analyses (focusing predominantly on Cochrane Meta-analyses when possible)”

Also, the description of different articles and the methodology used in each one is not present at all. The tools used in these articles to assess the outcome of their approach used for tinnitus management is also not present.

Answer: We agree with the reviewer, that a systematic review of therapeutic interventions in tinnitus should include comprehensive descriptions of the methodology (inclusion and exclusion criteria of studies), tools for outcome measurement and a detailed description of the available clinical trials. However, the purpose of this paper is not a systematic review of treatment interventions. We aim to discuss the current adaption of evidence based medicine in tinnitus.

Pages 4- 5, lines 143-189: too much information that need abstracting.

Answer: As mentioned above the purpose of the perspective paper is a discussion about the adaptation of evidence based medicine in tinnitus. Therefore we think that it is appropriate to describe the methodology for guideline development in detail.

Box 1 needs to be rearranged again. The methodologists used in RCTs mentioned in this box (such as physiotherapy and virtual reality based approaches) were not mentioned and the results sections at all.

Answer: Box 1 is intended to provide an overview about the variety of interventions that have been investigated for tinnitus treatment. It goes beyond the scope of this paper to describe the respective clinical trials in detail. According to the suggestion of the reviewer we added physiotherapy and VR based approaches in the result section (lines 243-247 and  lines 292-296)

“Virtual reality based treatment

In a large randomized clinical trial patients were randomized into either a virtual reality (VR) based intervention, or in CBT with both groups demonstrating similar improvement. These findings suggest the potential of VR based interventions and warrant further research.”

“Physiotherapy

Different forms of physiotherapy have been investigated with promising results. The majority of these studies focused on patients with tinnitus and comorbid temporomandibular joint or neck disorders. Large scale randomized controlled trials are needed to determine which interventions are effective in which patient groups.”

The authors mentioned some approaches used in tinnitus management such as tinnitus counselling, cognitive behavioral therapy, mindfulness and tinnitus… etc. Each one of these approaches should discussed in relation to the number of studies, the indication, the methodology, the results, and the method of qualification the outcome.

Answer: We agree with the reviewer, that a systematic review of therapeutic interventions in tinnitus should include detailed information about the number of studies, the indication, the methodology, the results, and the method of outcome measurement. However, the purpose of this paper is not a systematic review of treatment interventions, and it would go far beyond the scope of this paper to add all this information for the various interventions.

Discussion: Need to be more systematized starting with revising the tinnitus management approach followed by discussion of different guidelines and their effect sizes, and recommendation to be used or not. Consideration for future directions and conclusion are lengthy and need to be rewritten again and more comprehensive shorter statements.

Answer: The purpose of this paper is not a systematic review of the current tinnitus management approach.  The purpose of this perspective paper is the discussion of the EBM approach to tinnitus. Therefore we focused the discussion on the transition of clinical evidence in clinical guidelines and on suggestions to deal with the inherent challenges. To clarify this purpose we added the following sentence at the begin of the discussion (lines 415-417)

“We will first discuss the translation of the currently available clinical evidence into the different guidelines. In the second part we will provide suggestions for further development.”

Round 2

Reviewer 2 Report

Thank you for the revision of the manuscript. I have no more points.

Reviewer 3 Report

very redundant presentation of good work